# Armed with Faster Crypto: Optimizing Elliptic Curve Cryptography for ARM Processors

**DOI:** 10.3390/s24031030

**Published:** 2024-02-05

**Authors:** Ruben De Smet, Robrecht Blancquaert, Tom Godden, Kris Steenhaut, An Braeken

**Affiliations:** 1Department of Electronics and Informatics (ETRO), Vrije Universiteit Brussel, Pleinlaan 2, B-1050 Brussels, Belgium; rubedesm@vub.be; 2Department of Engineering Technology (INDI), Vrije Universiteit Brussel, Pleinlaan 2, B-1050 Brussels, Belgium; robrecht.simon.blancquaert@vub.be (R.B.); tom.godden@vub.be (T.G.); an.braeken@vub.be (A.B.)

**Keywords:** extended twisted Edwards curve, Curve25519, single instruction, multiple data (SIMD), Rust, ARM NEON

## Abstract

Elliptic curve cryptography is a widely deployed technology for securing digital communication. It is the basis of many cryptographic primitives such as key agreement protocols, digital signatures, and zero-knowledge proofs. Fast elliptic curve cryptography relies on heavily optimised modular arithmetic operations, which are often tailored to specific micro-architectures. In this article, we study and evaluate optimisations of the popular elliptic curve Curve25519 for ARM processors. We specifically target the ARM NEON single instruction, multiple data (SIMD) architecture, which is a popular architecture for modern smartphones. We introduce a novel representation for 128-bit NEON SIMD vectors, optimised for SIMD parallelisation, to accelerate elliptic curve operations significantly. Leveraging this representation, we implement an extended twisted Edwards curve Curve25519 back-end within the popular Rust library “curve25519-dalek”. We extensively evaluate our implementation across multiple ARM devices using both cryptographic benchmarks and the benchmark suite available for the Signal protocol. Our findings demonstrate a substantial back-end speed-up of at least 20% for ARM NEON, along with a noteworthy speed improvement of at least 15% for benchmarked Signal functions.

## 1. Introduction

Elliptic curve cryptography (ECC) has been deployed as the main asymmetric cryptographic primitive to secure various systems. Notably, transport layer security (TLS) 1.3 [1] added elliptic curve Diffie–Hellman key exchange (ECDHE) in the base specification, and it removed RSA-based cipher suites. Secure messaging applications like Signal and WhatsApp rely on ECC for their end-to-end encryption (E2EE) protocols. Signal additionally makes heavy use of ECC in its private group system [2] and its sealed unidentified delivery system [3], which greatly improves the security for Signal users.

ECC is appealing in those settings because of its unmatched performance and security properties. An *n*-bit secure elliptic curve key can be encoded in 2n+x bits, with *x* being a small number. This means that for the typical 128-bit security level, a key can be encoded in only 32 bytes.

Even considering the high potential throughput of ECC operations, a carefully optimised implementation of ECC algorithms is beneficial. In some applications—for example, zero-knowledge proofs (e.g., Bulletproofs [4])—thousands of ECC operations need to be carried out as efficiently as possible. Other applications might run on battery-powered devices and benefit from consuming as little power as possible for executing these computations.

In this article, we aim to improve the performance of the extended twisted Edwards curve of Curve25519 [5]. Curve25519 is one of the fastest curves in the literature. Since it is not covered by any patent [5], it has quickly gained popularity in many different applications. This article proposes a highly efficient implementation of Curve25519 elliptic curve operations for ARM processors since they are by far the most widely used processors in smartphone and tablet devices. Our proposed improvements help compensate for the limited processing power of such handheld devices. We specifically target ARM processors that support the ARM NEON single instruction, multiple data (SIMD) instruction set. We show an improvement in performance of 20%.

The main contributions in this article are the following:We present an adaption of the parallel formulae for extended twisted Edwards curves by Hisil et al. [6] for ARM processors by designing a new representation for an elliptic curve point using ARM NEON vectors.We provide an open source implementation of those formulae in the Dalek cryptography Curve25519 Rust library [7,8].We perform a performance analysis of our implementation on multiple ARM devices, demonstrating a significant speed-up.

## 2. Related Work

ECC is a popular technique for securing digital communication [9]. It is used in a range of cryptographic primitives, such as key agreement protocols, signatures, and zero-knowledge proofs (ZKPs) [10]. One of the key challenges in implementing ECC is the implementation of fast modular arithmetic operations, which can be optimised for specific microarchitectures [11].

Several articles have investigated the optimisation of ECC for specific hardware platforms. Our work builds on the previous efforts of Bernstein [5], Bernstein and Schwabe [9], and Hisil et al. [6]. Bernstein and Schwabe [9] describe the use of NEON vector instructions on ARM processors. Our library utilises the methods for multiplication described by Bernstein and Schwabe [9] for multiplication, reduction, and squaring within Curve25519. Instead of pure assembly, it uses a mix of Rust and assembly instructions and takes into account our new representation. Hisil et al. [6] propose a new addition algorithm for extended twisted Edwards curves that can be computed using four processors for an even faster implementation. We use this algorithm with Curve25519 [5] with our new representation for elliptic curve points.

Notable parallels can be drawn between our efforts and those outlined in related studies, although there are some key differences. Hamburg [12] introduced a swift and resource-efficient implementation of elliptic curve cryptography (ECC), albeit lacking ARM NEON instructions. Faz-Hernández et al. [13] harnessed AVX2 SIMD vector instructions, yielding substantial performance enhancements for ECC on contemporary Intel x86-64 architectures. In a similar vein, Cheng et al. [14] introduced a throughput-optimised AVX2 implementation of variable-base scalar multiplication. It is important to note that none of these references involve the utilisation of the parallel formulae of Hisil et al. [6].

Goetschmann et al. [15] leveraged Intel Skylake floating-point arithmetic to expedite elliptic curve algorithms, although without SIMD instructions. Our approach, in contrast, relies on integer arithmetic. Meanwhile, Dong et al. [16] achieved performance gains through an alternative avenue: harnessing embedded graphical processing units (GPUs).

Another body of research has concentrated on optimising ECC specifically for ARM processor architectures. Luc et al. [17] devised a technique to enhance point arithmetic efficiency on elliptic curves using ARM processors and NEON instructions. Additionally, Longa [18] introduced FourQNEON: an accelerated ECC scalar multiplication algorithm tailored for ARM processors. Our work differs from these previous optimisations for ARM NEON in that we create a new representation specifically for the parallel formulae of Hisil et al. [6] that works for ARM NEON.

An overview of these related works is presented in Table 1.

We target to improve the speed of the ECC algorithms using NEON, specifically by using the parallel formula for extended twisted Edwards curves proposed by Hisil et al. [6] relying on an ARM NEON SIMD representation. We do this by implementing a Curve25519 back-end in a widely used Rust library, curve25519-dalek [7], targeting the ARM NEON SIMD architecture. We evaluate the performance on a range of ARM devices and compare our results to the state-of-the-art implementation without SIMD.

## 3. Preliminaries

In this section, we first explain the different elliptic curve operations in the context of extended twisted Edwards curves. We then elaborate on how SIMD operations are currently implemented in the curve25519-dalek library for these operations. In Section 4, we build on this knowledge for speeding up the ECC operations, both using SIMD in a general way and specifically for ARM NEON.

### 3.1. Elliptic Curve Operations

The security of elliptic curve cryptography is based on the discrete logarithm assumption. This assumption states that multiplication of a point *P* on the elliptic curve by a scalar *a*, where a·P=A, is easy, but if knowing only *A* and *P*, it is generally very hard to find *a* in polynomial time. This problem is known as the (elliptic curve) discrete logarithm problem, and it allows elliptic curves to be used in cryptography as a public key cryptosystem.

Elliptic curves are a group of points that satisfy a specific equation: most commonly, the simplified Weierstrass equation $y^{2}=x^{3}+ax+b$. For any two points, we can create their sum by drawing a line through them and mirroring along the *x*-axis where the curve and the line intersect in a third point (by negating the *y* coordinate). This is illustrated in Figure 1. This figure also shows the addition of a point with itself, in which case the tangent along the curve is used as the line; this operation is called point doubling. Note that Figure 1 shows a continuous curve, whereas for cryptographic purposes, curves are defined over a finite field, but the properties for addition still hold.

From this addition, we can define multiplication of a point by a scalar as a series of additions and doublings: for example, 5P=2(2P)+P. Conforming to the geometric construction of the addition and doubling, the following are the formulae to perform these operations:$$\begin{array}{cccc} \; & \mathrm{Addition}: & \left(x_{3},y_{3}\right) & =\left({\left(\frac{y_{2}-y_{1}}{x_{2}-x_{1}}\right)}^{2}-x_{1}-x_{2},\left(\frac{y_{2}-y_{1}}{x_{2}-x_{1}}\right)\left(x_{1}-x_{3}\right)-y_{1}\right)\\ \; & \mathrm{Doubling}: & \left(x_{3},y_{3}\right) & =\left({\left(\frac{3x_{1}^{2}+a}{2y_{1}}\right)}^{2}-2x_{1},\left(\frac{3x_{1}^{2}+a}{2y_{1}}\right)\left(x_{1}-x_{3}\right)-y_{1}\right) \end{array}$$

### 3.2. Elliptic Curve Models and Point Representation

The above simplified Weierstrass equation and addition and doubling formulae have several shortcomings in the context of cryptography. First, the addition and doubling formulae are different. This leads to vulnerabilities such as side channel attacks [19], as by knowing the series of additions and doublings that was used, an attacker can uniquely restructure and derive the original scalar. The formulae given above are also lengthy, which is a disadvantage when they need to be repeated often. Researchers have looked for faster formulae that use fewer operations to achieve the same result. This has led to alternative curve representations, such as extended twisted Edwards curves [6], for which the curve equation is $ax^{2}+y^{2}=1+dx^{2}y^{2}$. The points on this curve consist of four coordinates, (X,Y,T,Z), and there is a single formula for doubling and addition:$$\left(x_{3},y_{3}\right)=\left(\frac{x_{1}y_{2}+y_{1}x_{2}}{1+dx_{1}x_{2}y_{1}y_{2}},\frac{y_{1}y_{2}-ax_{1}x_{2}}{1-dx_{1}x_{2}y_{1}y_{2}}\right)$$

With this representation, more coordinates per point need to be tracked, but operations are faster as there are fewer field operations in the formula. However, a new problem is introduced by the fact that the order of extended twisted Edwards curves is not prime. These curves generally have a cofactor of 8, which means the order of the curve is 8·p, where *p* is a large prime. This cofactor can lead to vulnerabilities such as small-subgroup attacks, which should be taken into account [20]. These can be mitigated by ensuring that the points used in cryptographic operations are in the correct subgroup. This can be done using a costly multiplication or by using optimised methods such as Ristretto [21], which is based on the Decaf method of Hamburg [20]. The advantages of the speed-up generally outweigh the drawbacks of cofactor elimination methods, especially since the extended twisted Edwards coordinates have additional optimisations using parallel computation.

The extended twisted Edwards addition formula for $\left(X_{1}:Y_{1}:T_{1}:Z_{1}\right)$ + $\left(X_{2}:Y_{2}:T_{2}:Z_{2}\right)$ = $\left(X_{3}:Y_{3}:T_{3}:Z_{3}\right)$ can be written as the parallel formula seen in Table 2 using four processors (*k* is a predetermined constant) as formulated by Hisil et al. [6]:

In the cost column, *M* represents a multiplication in the field, and *D* is a multiplication by a known scalar. This algorithm describes a way to add two curve points in only five steps and with a cost of 2M+1D. In reality, the algorithm cannot be split over four separate processing units due to the need to synchronise after every step, which introduces too large an overhead. Fortunately, SIMD instructions provide a way to execute this optimised algorithm. We are able to represent each column in the parallel formula above as a sub-vector in a larger SIMD vector. On each of these sub-vectors, the same operation can be executed simultaneously, such as addition or multiplication with another SIMD vector or multiplication with a constant. This operation happens pair-wise, i.e., the first sub-vector of the first SIMD vector will be added to the first sub-vector of the second SIMD vector.

### 3.3. Curve25519 and Field Polynomials

Curve25519 [5] was specifically designed to be fast by having efficient formulae. At the same time, Curve25519 avoids many implementation-related vulnerabilities. Such vulnerabilities include side-channel attacks and certain algebraic attacks. This specific design of the curve avoids some of these side-channel attacks, as described by Fan et al. [22] and Abarzúa et al. [23]. Countermeasures against other side-channel attacks still rely on the implementation. Avoiding these other side-channel attacks is out of the scope of this paper. However, the implementation described in this paper is based on an already existing popular implementation of Curve25519 that has no known vulnerabilities. It uses algorithms that are known to be safe against timing-based attacks. Our implementation, which adapts these algorithms, should, in theory, have the same safety.

The speed and security of Curve25519 derive from specifically chosen properties such as the field size and the fixed-base point that allow for efficient arithmetic operations while remaining secure. Curve25519 is a curve over the finite prime field of order $2^{255}-19$ with the following Weierstrass equation: $y^{2}=x^{3}+486662x^{2}+x$, which translates into the extended twisted Edwards equation of: $-x^{2}+y^{2}=1-\frac{121665}{121666}x^{2}y^{2}$.

Operations in the field of this elliptic curve are done with 255-bit numbers. To represent them inside our algorithm, we can split these numbers into 10 numbers with radixes of $2^{25.5}$. In reality, these will be alternatively 25- and 26-bit numbers and radixes. Since CPU representations that contain these numbers will be at least 32 bits, there are some unused bits after each number. These are useful to ensure that algorithms need to do fewer reductions to field size. This leads to an increase in performance compared to an implementation with a larger radix and more reductions.

The polynomial representation for a field element *x* takes the form of $2^{0}x_{0}+2^{25.5}x_{1}+2^{51}x_{2}+\ldots +2^{229.5}x_{9}$, where each component of *x* is referred to as a *limb*. This representation illustrates how addition and multiplication operations should be performed when the field element is divided into limbs, as it is equivalent to performing these operations on the polynomials.

When two of these polynomials are multiplied together, the resulting polynomial will contain terms from $2^{0}x_{0}$ to $2^{229.5}x_{9}$ but also additional terms ranging from $2^{255}x_{10}$ to $2^{459}x_{18}$. The $x_{10}$ to $x_{18}$ terms can be reduced by multiplying the resulting term with $19\;\cdot \;2^{-255}$. This follows from:$$2^{255}-19\equiv 0\;\mathrm{mod}\;2^{255}-19\Leftrightarrow 2^{255}\equiv 19\;\mathrm{mod}\;2^{255}-19$$

Thus:$$2^{255}x_{10}\equiv 19\;\cdot \;2^{0}\;\cdot \;x_{10}\;\mathrm{mod}\;2^{255}-19$$

This effectively reduces the *x*-index of these terms by 10, so the term of $x_{10}$ is added to the term of $x_{0}$, $x_{11}$ to $x_{1}$, … This representation and these reduction techniques are exactly how we will represent and reduce field elements using SIMD. To add two polynomials together, terms of the same power are added together, whereas in a SIMD addition, sub-vectors with the same index in two SIMD vectors are added together.

## 4. Techniques for Speed-Up with SIMD

### 4.1. SIMD in Curve25519-Dalek

SIMD allows us to perform a single instruction on multiple elements in a SIMD vector at the same time. This is applied in the curve255119-dalek library with the parallel extended twisted Edwards formulae for the AVX2 architecture. We implement Hisil et al. [6] for ARM NEON, creating a new representation accounting for the 128-bit SIMD vector size. We will first discuss how elliptic curve points and field elements are represented in the library.

In Section 3, we discussed how a field element is split into ten pieces of 32 bits with some unused bits per limb. These unused bits are used to extend the bound of the field order so that after some operations, such as addition, these bits are used instead of performing a reduction step. Using these bits, we can delay the costly reduction step as long as possible. This distribution is represented in Figure 2.

Every limb is not immediately put in the first open place in a vector. This follows from the fact that in the extended twisted Edwards representation, every point is constructed of four field elements and that we have to distribute the ten limbs of each of those four elements over the SIMD vectors. First, we will consider the already implemented case in the existing curve25519-library, for which these SIMD vectors are 256 bits long in the AVX2 architecture. These 256-bit vectors can only hold 8 limbs, so we will need 5 vectors to hold all 40 limbs of the four field elements of an elliptic curve point projective representation. If we have four field elements (A,B,C,D), we can divide each element *X* into limbs $x_{0}$ to $x_{9}$. This results in the structure shown in Figure 3, where each row represents a SIMD vector consisting of the limbs (A,B,C,D). In this representation, we call each column a *lane*.

This table of five 256-bit SIMD vectors now holds all the information about a single elliptic curve point. The limbs ${\left\{a,b,c,d\right\}}_{0,\cdots ,9}$ are distributed in the vectors $v_{i}$ as follows:$$i\in \left\{0,1,2,3,4\right\},n=2i:v_{i}=\left(a_{n},b_{n},a_{n+1},b_{n+1},c_{n},d_{n},c_{n+1},d_{n+1}\right)$$

This representation makes multiplication between elliptic point tables easier. When we multiply two 32-bit numbers together, the result will be a 64-bit number. Because our SIMD operations happen on the whole vector at the same time, a multiplication would be impossible without losing information when using the default above representation because there is not enough space. However, this representation allows us to easily double the number of vectors by extracting the *b* and *d* lanes and bit-shifting by 32 to the left, as can be seen in Figure 4. In other words, we take out all the *b* and *d* limbs, substitute them with 0, place them in a new vector in the same place as the *a* and *c* limbs would be, and fill the rest with 0 again.

After a multiplication of two limbs, those zeroed sub-vectors will be filled with the higher 32 bits of the 64-bit result of the multiplication. This 64-bit result can then be reduced modulo the field order, which reduces each limb once again to 32 bits, and put back in the default representation by interweaving the vectors: effectively the reverse of the operations seen in Figure 4.

### 4.2. Implementation on ARM NEON

For our representation using ARM NEON, we only have vectors of 128 bits to work with. To keep using the same techniques, we have split the representation of a point into two tables.

This would theoretically double the number of operations in algorithms compared to an implementation without this split. An elliptic curve point representation previously consisting of five SIMD vectors will now use ten, and since each operation on such a representation applies over all SIMD vectors, the number of operations is doubled.

There are some cases in which we can make use of the split of tables without drawbacks—or even to our advantage. For example, switching the *A* and *B* lanes with the *C* and *D* lanes is equivalent to trivially swapping the two variables holding each table in memory. This is significantly simpler and more efficient than the equivalent SIMD operation to rearrange these vectors. This swapping of lanes is necessary when an operation needs to happen between two different lanes of two elliptic curve point tables, e.g., the *A* and *D* lanes. When we want to add what is in the *A* lane of one representation to what is in the *D* lane of the other, we would have to place the *A* lane in the *D* lane or vice versa. Operations such as addition happen pair-wise between lanes of two representations—the *A* lane of the first gets added to the *A* lane of the second—so if we want to add a different lane, we need to swap it first.

This swapping and other techniques we used for optimisation will be further discussed in Section 4.3. First, we will present the functions in which our back-end is used and how it is used.

Our goal when speeding up ECC is to make the scalar multiplication faster. This scalar multiplication is done using an algorithm that utilises the addition algorithm between two elliptic curve points. This addition algorithm uses our back-end. The addition algorithm of the curve25519-dalek [7] library adds a CachedPoint to an ExtendedPoint. An ExtendedPoint is the elliptic curve point representation as discussed above. A CachedPoint is the same, but it has some pre-computed variables. This follows from the parallel formula seen in Table 2, where the first steps only use the elements of one elliptic curve point at a time; thus, when using the same elliptic curve point multiple times for addition in a row, it is more efficient to pre-compute this step.

To create a CachedPoint from a point $\left(X_{1},Y_{1},Z_{1},T_{1}\right)$, we execute the following steps Algorithm 1. The value 121666 in this algorithm is the constant *k* in the parallel formula.
**Algorithm 1** Algorithm for creation of a CachedPoint 1: **function** from($P_{1}$: ExtendedPoint) 2:      $\left(X_{1},Y_{1},Z_{1},T_{1}\right)\leftarrow P_{1}$ 3:      $\left(S_{2},S_{3},Z_{1},T_{1}\right)\leftarrow \left(Y_{1}-X_{1},Y_{1}+X_{1},Z_{1},T_{1}\right)$ 4:      $\left(S_{2}^{\prime },S_{3}^{\prime },Z_{1}^{\prime },T_{1}^{\prime }\right)\leftarrow \left(121666\;\cdot \;S_{1},121666\;\cdot \;S_{2},2\;\cdot \;121666\;\cdot \;Z_{1},2\;\cdot \;121666\;\cdot \;T_{1}\right)$ 5:      $P_{1}^{\prime }\leftarrow \left(S_{2}^{\prime },S_{3}^{\prime },Z_{1}^{\prime },T_{1}^{\prime }\right)$ 6:      **return** $P_{1}^{\prime }$: CachedPoint 7: **end function**

Step 2 gives us a CachedPoint. To add two elliptic curve points, $P_{1}=\left(X_{1},Y_{1},Z_{1},T_{1}\right)$ and $P_{2}=\left(X_{2},Y_{2},Z_{2},T_{2}\right)$, we first transform $P_{2}$ into a CachedPoint: $P_{2}^{\prime }=\left(S_{2}^{\prime },S_{3}^{\prime },Z_{2}^{\prime },T_{2}^{\prime }\right)$. Then, we follow the steps in Algorithm 2 to perform the elliptic curve point addition of $P_{1}+P_{2}^{\prime }=P_{3}$.

This yields the same results as the parallel formula of Hisil et al. [6], with a slightly different execution order due to the CachedPoint. Some steps are omitted in the above algorithms that are necessary when executing them with SIMD vectors. We treat each element in a SIMD vector, such as (X1,Y1,Z1,T1), as a separate variable that we can move and manipulate. In reality, these are all stored in a singular SIMD vector, and moving and performing operations on them is a bit more involved. For example, on Line 3 of Algorithm 1 and Line 4 of Algorithm 2, we perform the operation (Y1−X1,Y1+X1,Z1,T1). All the separate steps required for this operation with SIMD are given in Algorithm 3.
**Algorithm 2** Algorithm for adding ExtendedPoint to CachedPoint 1: **function** add(P1: ExtendedPoint, P2′: CachedPoint) 2:      (X1,Y1,Z1,T1)←P1 3:      (S2′,S3′,Z2′,T2′)←P2′ 4:      (S0,S1,Z1,T1)←(Y1−X1,Y1+X1,Z1,T1) 5:      (S8,S9,S10,S11)←(S0·S2′,S1·S3′,Z1·Z2′,T1·T2′) 6:      (S12,S13,S14,S15)←(S9−S8,S9+S8,S10−S11,S10+S11) 7:      (X3,Y3,Z3,T3)←(S12·S14,S15·S13,S15·S14,S12·S13) 8:      P3←(X3,Y3,Z3,T3) 9:      **return** P3: ExtendedPoint10: **end function**

**Algorithm 3** SIMD instructions necessary to calculate $\left(Y_{1}-X_{1},Y_{1}+X_{1},Z_{1},T_{1}\right)$ from $\left(X_{1},Y_{1},Z_{1},T_{1}\right)$ 1: $\left(X_{1},Y_{1},Z_{1},T_{1}\right)\leftarrow P_{1}$ 2: ${\mathrm{temp}}_{1}\leftarrow \left(Y_{1},X_{1},T_{1},Z_{1}\right)\leftarrow$ Shuffle($P_{1}$, BADC) 3: ${\mathrm{temp}}_{2}\leftarrow \left(-X_{1},-Y_{1},-Z_{1},-T_{1}\right)\leftarrow$ Negate($P_{1}$) 4: ${\mathrm{temp}}_{2}\leftarrow \left(-X_{1},Y_{1},-Z_{1},T_{1}\right)\leftarrow$ Blend($P_{1}$, ${\mathrm{temp}}_{1}$, AC) 5: $\mathrm{result}\leftarrow (Y_{1}-X_{1},X_{1}+Y_{1},T_{1}-Z_{1},Z_{1}+T_{1})\leftarrow$ Add(${\mathrm{temp}}_{1}$, ${\mathrm{temp}}_{2}$) 6: $(Y_{1}-X_{1},X_{1}+Y_{1},Z_{1},T_{1})\leftarrow$ Blend($P_{1}$, result, AB)

In Algorithm 3, we use functions in Lines 2–5 to perform operations on the SIMD vector. These are the functions in which we use ARM NEON intrinsics to optimise performance. They are further explained in Section 4.3.

### 4.3. Techniques Used for Optimisation

We have optimised functions such as “shuffle”, “negate”, “blend”, and “add” seen in Algorithm 3 for ARM NEON. These optimisations are based on the adaptation of the the back-end for the Intel AVX2 SIMD vector instructions such that it can use our ARM NEON representation and specific instructions. They all happen in the *field* of the elliptic curve: e.g., adding two of our representations means adding the sub-vectors pair-wise and not performing an elliptic curve point addition. Below, we will discuss these field algorithms.

#### 4.3.1. Shuffle

The shuffle function takes an input set of field elements *ABCD* and returns a new sequence of lanes according to a control sequence: for example, *AAAA* or *BADC*. The problem with our representation of a two SIMD vector solution is that lanes that are stored in the first vector might need to be swapped with lanes in the second vector. To account for this, we give both SIMD vectors of our representation to the shuffle! macro of the Rust packed-simd crate [24]. We do this shuffle! once for each SIMD vector, so twice in total. We could use the vqtbx1q_u8 ARM NEON intrinsic, which can also combine two SIMD vectors by reordering every 8 bits according to a third input vector; however, this instruction is slow according to the specification. This instruction would only be preferable if it is necessary to reorder every sub-vector of 8 bits instead of the sub-vectors of 32 bits we work with. Instead, it is better to use multiple other instructions to get the same result.

This is what the shuffle! macro does. It first lowers to the shufflevector LLVM instruction, which becomes a sequence of assembly instructions using ARM NEON intrinsics such as trn1 and trn2 to get the desired reordering of the vectors. The instructions trn1 and trn2 combine vectors by taking, respectively, the even- or odd-numbered sub-vectors from the first input vector and vice versa for the second input vector. This, combined with some instructions to extract and insert sub-vectors from the SIMD vector, gives us our wanted output.

#### 4.3.2. Blend

Blend behaves in much the same way as shuffle but merges two field elements together based on an input. For example, given two field elements $A_{1}B_{1}C_{1}D_{1}$ and $A_{2}B_{2}C_{2}D_{2}$ and an input lane C, blend returns $A_{1}B_{1}C_{2}D_{1}$ as the C input as dictated by the C lane being taken from the second input field element.

This blend function is performed using the shuffle! macro with some optimisations. A naive implementation would use shuffle! twice to combine the first SIMD vectors of the inputs and then again for the second vectors. However, this is not always necessary. When taking everything from the first SIMD vector from the first input and everything from the second SIMD vector from the second input or vice versa, we can simply take those SIMD vectors as our output without having to perform a shuffle. Similarly, when taking only one lane from the first or second input, only the SIMD vector holding that lane has to be shuffled, and we can directly take the other vector.

This input that decides how to blend the lanes or how to reorder in the shuffle function is determined by the formula in which it is used. Thus, it is always known at compile time and does not raise issues of not being constant time as it does not depend on the input of which points are used with the algorithm.

#### 4.3.3. Negate

To negate within the finite field, we subtract the field element from a multiple of the field order. The multiple of the field order is taken to avoid an underflow as −x≡k·p−xmodp for any integer *k* with field order *p*. In certain algorithms when we know the bounds are low, we can perform a lazy negation with 2·p and without a reduction, ensuring that the bound stays low. Otherwise, if the bounds are high, we can perform a reduction with 16·p and perform a reduction afterwards. This still requires knowing the bounds beforehand, as we cannot exceed the field size. This implementation is equivalent to the AVX2 implementation.

#### 4.3.4. Unpack and Repack

In order to multiply a SIMD vector with another SIMD vector or constant, it first has to be *unpacked*. This is done by splitting each vector up into two, with each vector taking every other limb, as seen in Figure 5. Then, after multiplication, the vector has to be *repacked*, which combines two vectors into one.

There is a key difference here with the AVX2 implementation. AVX2 splits into two SIMD vectors for multiplication as described in Figure 5. For ARM NEON however, the multiplication SIMD instructions expect two 64-bit vectors that result in a 128-bit vector. The SIMD vector is still split in two, with each result vector taking every other limb, but the result vectors are of length 64 bits. This splitting can be seen in Figure 6, and our representation using two ARM NEON SIMD vectors results in four SIMD vectors. We create these four vectors using the vget_low_u32 and vget_high_u32 ARM NEON instructions, which, respectively, get the lower two and the higher two limbs from a SIMD vector.

After multiplication, a reduction always happens. We explain reduction in more detail in Section 4.3.5. This reduced form will consist of alternating limbs and zeroed sub-vectors, as described in Figure 5. Repacking the vectors into the default representation involves extracting the limbs and putting them into new SIMD vectors. We do this by extracting $\left(a_{0},0\right)$ with vget_low_u32 and then inserting $b_{0}$ with vset_lane_u32 to get $\left(a_{0},b_{0}\right)$. The $b_{0}$ is first extracted with the vgetq_lane_u32, which can extract an arbitrary limb from a SIMD vector. In the same way, we obtain $\left(a_{1},b_{1}\right)$ to combine both using vcombine_u32 and get $\left(a_{0},b_{0},a_{1},b_{1}\right)$. This process is repeated again to obtain our second vector $\left(c_{0},d_{0},c_{1},d_{1}\right)$.

#### 4.3.5. Reduce and Reduce64

The reduce function is called to reduce an elliptic curve point field element in our default two SIMD vector representation. A reduction is performed by adding the extra bound bits of each limb to the next limb, and for the last limb, we add it to the first after a multiplication with 19 to conform with the modulo operations. The same is done with reduce64, except the extra bits are now 25 or 26 bits long after a multiplication.

These functions are implemented similarly to the ones in the existing AVX2 implementation, except that, similar to functions described above, extraction from vectors and combination of vectors is done with vget_high_u32, vget_low_u32, and vcombine_u32.

#### 4.3.6. Operations on Elliptic Curve Point Representations

The add, multiplication with scalar, and multiplication between field element operations are again a straightforward reimplementation of the AVX2 functions ported to ARM NEON intrinsics.

Addition is a straightforward operation, for which we again rely on the packed_simd_2 crate to provide the optimal instructions. When adding two elliptic curve point representations together vectorwise, we simply add the corresponding vectors of the first point to the second point.

For multiplication by a scalar, each SIMD vector in our representation is multiplied by a scalar using the vmull_u32 intrinsic. As these scalars can be large, the representation is first unpacked, then the multiplication happens, and then the result is reduced with reduce64 before being repacked.

The same unpacking, multiplication, reduction, and repacking happens for the multiplication between the elliptic curve point representation. The algorithm for this is a direct reimplementation of the formula described in *Multiplication mod $2^{255}$ on NEON* by Bernstein and Schwabe [9].

## 5. Performance Analysis Method

Our optimised ARM NEON code was evaluated on several devices, as shown in Table 3. The same benchmarks were run on all the devices, with code compiled specifically for the architecture of each device. These benchmarks were compiled targeting either the non-SIMD back-end, the non-SIMD back-end but using auto-vectorisation, or our new NEON SIMD back-end. With auto-vectorisation, the compiler will try to automatically use SIMD instructions to improve performance.

The benchmarks use the criterion crate [25] to measure the time it takes to execute specific functions. Each function benchmark executes the function under study repeatedly until a preset time has elapsed: usually 5 s. Performance measurements start after a warm-up phase, which gives time for the OS and CPU to adapt to the workload. We give the 95% confidence interval for the median, calculated using linear regression. We also give the measured median value. Speed-up is given as the difference between median measured values, with statistical significance calculated using a T-test to always have a *p* value lower than 0.001.

The dalek-curve25519 library provides many benchmarks; below, we will discuss the results for the two most relevant ones. These are: a function that either uses the SIMD or the serial back-end to multiply a field element by a scalar (*constant-time variable-base scalar multiplication*) and a practical use case in the Signal library to decrypt a universally unique identifier (*Decrypt UUID*).

## 6. Results

The results of our benchmarks are discussed in this section. For each of our evaluations, we show a table with the summarised results of the benchmark as well as a figure that depicts the results of the benchmarks with a violin plot.

### 6.1. Constant-Time Variable-Base Scalar Multiplication

Table 4 displays the results of the benchmarks of constant-time variable-base scalar multiplication. This shows little to no improvement from the non-SIMD version to the auto-vectorised. From non-SIMD to the NEON version, there is an improvement on Jolla, Pi, and X10 II of, respectively, 53.47%, 24.74%, and 21.22%. Figure 7 displays a violin-plot of the results.

### 6.2. Decrypt UUID

Similar to the previous point, the results of the benchmark of the decrypt UUID function can be found in Table 5, with a violin plot of the results in Figure 8. This function aims to give a more practical example of the usage of the SIMD back-end in an actual library (Signal). Auto-vectorised again has little to no speed-up, while NEON SIMD performs 36.67%, 15.52%, and 13.05% faster than baseline on Jolla, Pi, and X10 II, respectively.

### 6.3. Discussion of Results

The results are completely in line with expectations. Since curve25519-dalek [7] reports a 40% speed-up for AVX2 and we use roughly twice the number of instructions, we get the expected speed-up of 20%. It seems that the speed-up is lower on newer devices, though more extensive testing on a variety of different cores, looking specifically at clock speeds, boost, architecture, etc. would be necessary to get a clear answer as to why. The benchmarks demonstrate the efficiency of using the four-way parallel formulae of Hisil et al. [6], even with smaller SIMD vectors. This indicates that formulae specifically designed for four-way parallelism, and thus for 256-bit SIMD architectures, can still work on 128-bit SIMD architectures. The inherent parallelism of SIMD provides a speed-up for these algorithms, even on smaller SIMD vector sizes.

Small further optimisations might be possible by changing some of the SIMD instructions used in the proposed implementation. However, given the use of a state-of-the-art algorithm for extended twisted Edwards curves and the fact that our speed-up already is in line with the theoretically expected speed-up, large improvements would require more fundamental changes to the elliptic curve algorithms themselves.

## 7. Future Work

Adapting the results of this paper to different architectures will still require some work, as the input and output of the SIMD instructions will differ. However, this adaptation should be moderately easy for 256-bit SIMD vectors such as the original implementation or 128-bit SIMD vectors such as our newly proposed implementation. Future work will focus on making our implementation more adaptable to different architectures and should give similar results.

For different SIMD vector sizes, a new implementation will be more difficult. Changes in the bounds and other parts of the algorithm, like the sequence of operations, will necessitate the use of other SIMD vector sizes. Future work should also explore how to adapt the linear algorithm that was originally used to create the four-way parallelism by Hisil et al. [6] to create new algorithms for different SIMD vector sizes. The same might also be done to create SIMD algorithms for other elliptic curve types.

## 8. Conclusions

As can be seen from Section 6, the possible speed-up from using an ARM NEON back-end to perform elliptic curve operations for extended twisted Edwards curves using SIMD is at least 20%. This speed-up varies greatly from device to device, but can be seen in every device and can reach as high as 50%. It is generally less than the 40% speed-up generated by the AVX2 version of the SIMD back-end, but that is to be expected, as NEON SIMD only has vectors of half the size. These results emphasise the possibility of SIMD to speed-up elliptic curve cryptography, even with relatively small SIMD vector sizes, by employing the parallel formulae by Hisil et al. [6]. This usage of parallel formulae is non-trivial and cannot be automated by auto-vectorisation at the moment. This paves the way to possibly employ similar parallel formulae for other elliptic curve models other than extended twisted Edwards curves and to generalise the algorithm to work for any SIMD vector size.

## Figures and Tables

**Figure 1 sensors-24-01030-f001:**
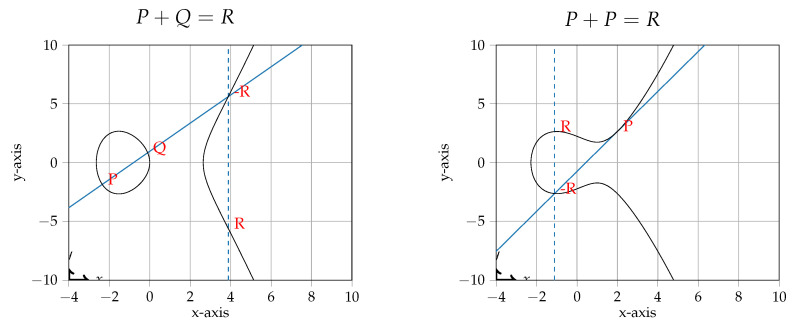
Addition of points on an elliptic curve. The full line indicates the drawn line to get −R. The dashed line indicates the reflection along the *x*-axis to get *R*.

**Figure 2 sensors-24-01030-f002:**
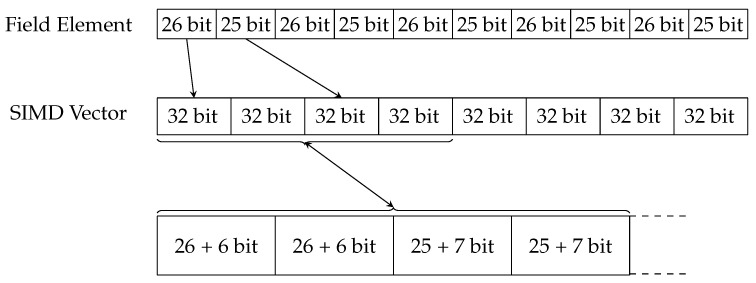
Distribution of one field element’s limbs over a vector: of the ten limbs of the first field element, two are put in specific sub-vectors in the first SIMD vector. These sub-vectors are made up of the limbs plus extra bits used to exceed the bound of the field order.

**Figure 3 sensors-24-01030-f003:**
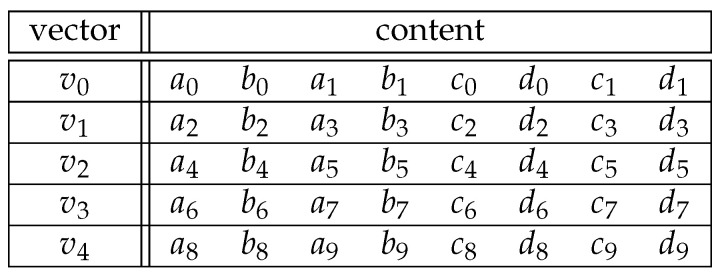
Distribution of field element limbs over vectors for the representation of an elliptic curve point. The first vector holds the first two limbs of field elements (A,B,C,D), the second vector the next two, and so on. The alternating limbs are distributed in such a way as to make operations such as multiplication easier.

**Figure 4 sensors-24-01030-f004:**
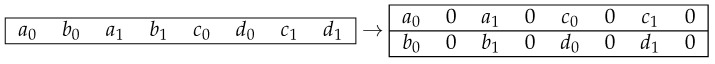
Split of field element limbs over vectors, which prepares for a multiplication.

**Figure 5 sensors-24-01030-f005:**
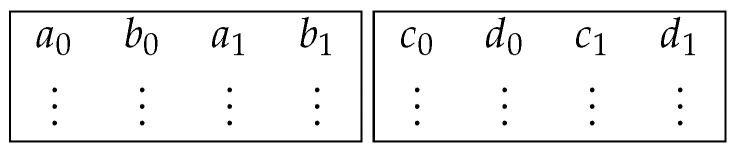
Splitting of vectors.

**Figure 6 sensors-24-01030-f006:**
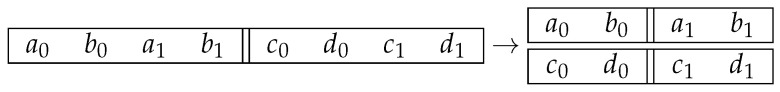
Split of 128-bit ARM NEON vector into 64-bit vectors in preparation for multiplication.

**Figure 7 sensors-24-01030-f007:**
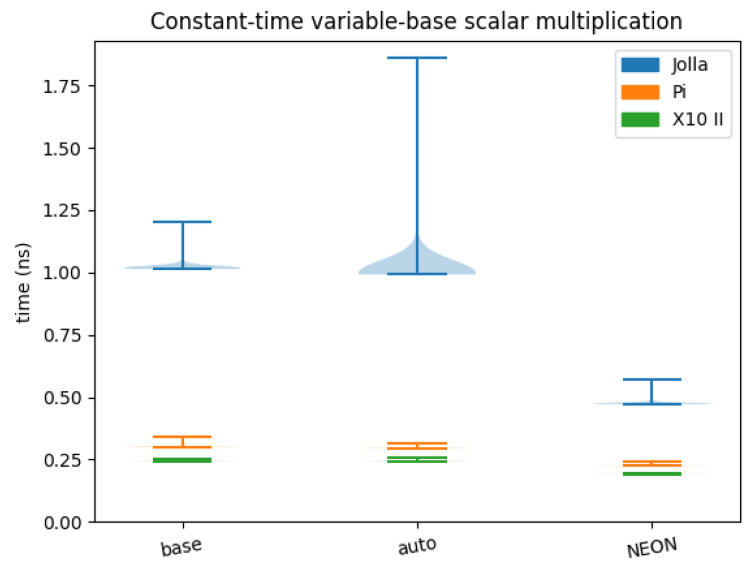
Constant-time variable-base scalar multiplication benchmarks.

**Figure 8 sensors-24-01030-f008:**
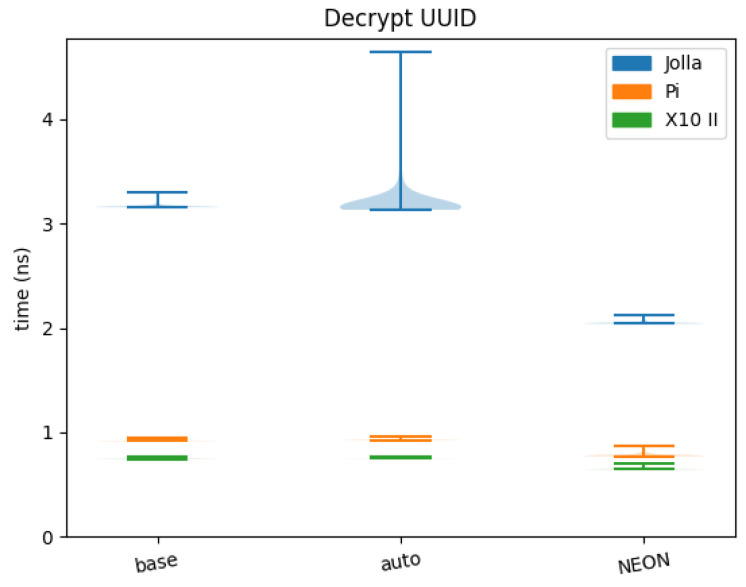
Decrypt UUID benchmarks.

**Table 1 sensors-24-01030-t001:** Related work, showing a general overview of the target elliptic curves and CPU architectures of related papers.

Title	Year	Curve	Target	SIMD
NEON Crypto [9]	2012	Multiple (including Curve25519)	ARM NEON	Yes
Twisted Edwards Curves Revisited [6]	2008	extended twisted Edwards curve	All	No
Fast and Compact Elliptic Curve Cryptography [12]	2012	Multiple types (Montgomery, twisted Edwards)	All	No
High-performance Implementation of Elliptic Curve Cryptography Using Vector Instructions [13]	2019	Ed25519, Ed448, Curve25519, Curve448	AVX2	Yes
High-Throughput Elliptic Curve Cryptography Using AVX2 Vector Instructions [14]	2021	Ed25519, Curve25519	AVX2	Yes
Fast implementations of Curve25519 on Intel Skylake [15]	2020	Curve25519	AVX2	Yes
EC-ECC: Accelerating Elliptic Curve Cryptography for Edge Computing on Embedded GPU TX2 [16]	2022	Ed25519, Ed448, Curve255519, Curve448	GPU	No
Improving the Efficiency of Point Arithmetic on Elliptic Curves Using ARM Processors and NEON [17]	2022	All	ARM NEON	Yes
FourQNEON: Faster Elliptic Curve Scalar Multiplications on ARM Processors [18]	2017	All	ARM NEON	Yes
Our work	2023	Curve25519 (extended twisted Edwards equivalent)	ARM NEON	Yes

**Table 2 sensors-24-01030-t002:** Parallel formula for elliptic point addition as formulated by Hisil et al. [6].

Cost	Step	Processor 1	Processor 2	Processor 3	Processor 4
	1	$R_{1}\leftarrow Y_{1}-X_{1}$	$R_{2}\leftarrow Y_{2}-X_{2}$	$R_{3}\leftarrow Y_{1}+X_{1}$	$R_{4}\leftarrow Y_{2}+X_{2}$
1*M*	2	$R_{5}\leftarrow R_{1}\;\cdot \;R_{2}$	$R_{6}\leftarrow R_{3}\;\cdot \;R_{4}$	$R_{7}\leftarrow T_{1}\;\cdot \;T_{2}$	$R_{8}\leftarrow Z_{1}+Z_{2}$
1*D*	3	idle	idle	$R_{7}\leftarrow kR_{7}$	$R_{8}\leftarrow 2R_{8}$
	4	$R_{1}\leftarrow R_{6}-R_{5}$	$R_{2}\leftarrow R_{8}-R_{7}$	$R_{3}\leftarrow R_{8}+R_{7}$	$R_{4}\leftarrow R_{6}+R_{5}$
1*M*	5	$X_{3}\leftarrow R_{1}\;\cdot \;R_{2}$	$Y_{3}\leftarrow R_{3}\;\cdot \;R_{4}$	$T_{3}\leftarrow R_{1}\;\cdot \;R_{4}$	$Z_{3}\leftarrow R_{2}\;\cdot \;R_{3}$

**Table 3 sensors-24-01030-t003:** Devices used for benchmarking. Two single-board computers (SBCs), four smartphones, and two Intel devices were used.

Device	CPU	Architecture	Identifier
Raspberry Pi 4Model B rev 1.4	Broadcom BCM2711	ARMv8 Cortex-A72	Pi
Jolla	Qualcomm Snapdragon 400 MSM8930	ARMv7 Cortex-A9	Jolla
Sony Xperia 10 II	Qualcomm Snapdragon 665	ARMv8 Cortex-A73 and A53	X10 II

**Table 4 sensors-24-01030-t004:** Benchmark results for constant-time variable-base scalar multiplication on various devices for various targets showing speed-up for NEON over base.

Device	Target	Median (μs)	95% CI (μs)			Speed-Up
Jolla	base	1017.3	1017.2	to	1017.6	base
Jolla	auto	995.6	995.4	to	995.8	*
Jolla	NEON	473.3	473.2	to	473.5	53.47%
Pi	base	299.3	299.3	to	299.5	base
Pi	auto	294.4	294.3	to	294.4	*
Pi	NEON	225.3	225.2	to	225.5	24.74%
X10 II	base	242.0	241.9	to	242.3	base
X10 II	auto	242.4	242.3	to	242.5	*
X10 II	NEON	190.7	190.6	to	190.8	21.22%

*: auto-vectorised speed-up results not displayed because they are not statistically significant or are very small.

**Table 5 sensors-24-01030-t005:** Benchmark results of decrypt UUID on various devices for various targets, showing speed-up for NEON over base.

Device	Target	Median (μs)	95% CI (μs)			Speed-Up
Jolla	base	3153.1	3153.1	to	3153.1	base
Jolla	auto	3120.3	3100.5	to	3122.1	*
Jolla	NEON	2028.4	2026.6	to	2044.9	36.67%
Pi	base	918.9	918.6	to	920.1	base
Pi	auto	922.5	922.1	to	922.9	*
Pi	NEON	776.3	775.6	to	777.5	15.52%
X10 II	base	747.3	747.1	to	747.4	base
X10 II	auto	751.3	751.2	to	751.5	*
X10 II	NEON	647.8	647.6	to	647.9	13.05%

*: auto-vectorised speed-up results not displayed because they are not statistically significant or are very small.

## Data Availability

Dataset available on request from the authors.

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
