# Peer review of "Armed with Faster Crypto: Optimizing Elliptic Curve Cryptography for ARM Processors"

_sensors, 2024, doi:10.3390/s24031030_

Round 1
Reviewer 1 Report
Comments and Suggestions for Authors
The present work proposes an optimization of elliptic curve cryptography for Arm processors. The authors begin with a brief description of the topic they have investigated, ending with the main contributions of the study. Next, based on solid and relevant references, they develop an extensive section of related works. Next, they make clear the different elliptic curve operations as well as the techniques for speed-up with SIMD and the entire method they used in the study. Finally, they show the results and conclusions.
Despite being a fairly theoretical work and it seems to be well modeled and structured, some circumstances can be seen that could be improved in order to be more useful for other research groups:
- The study does not make clear what would happen if this process were applied to another type of processor.
- One more extension should be made in the results section, they are very brief. The data that has resulted from the research could be related, asking several questions: Are they as expected? Could they improve with some alternative in the proposed algorithms?
- Additionally, information about possible future work should be added. Which would mean that other research groups, or the authors themselves, could continue the study soon.
Reviewer 2 Report
Comments and Suggestions for Authors
Review Report
Author’s contribution:
Armed with Faster Crypto:Optimizing Elliptic Curve Cryptography for Arm Processors
The authors focused on the ARM NEON simd architecture, a popular architecture for modern smartphones. They introduce a novel representation for 128-bit NEON SIMD vectors, optimised for SIMD parallelisation, to accelerate elliptic curve operations significantly. Leveraging this representation, an extended twisted Edwards curve Curve25519 back-end within the popular Rust library "curve25519-dalek is implemented.
Traditional Elliptic curve cryptography (ECC) has been deployed as the main asymmetric cryptographic primitive to secure various systems. The proposed method is reliable for Single instruction multiple data (SIMD).
The findings demonstrate a substantial back-end speed-up of at least 20% for ARM NEON, along with a noteworthy speed improvement of at least 15% for benchmarked Signal functions.
Comments and Suggestions:
1. This paper is well-written and well-organized.
2. The mathematical formulations are well laid out.
3. The results are presented clearly and support the authors’ hypothesis.
4. Conclusions are consistent with the evidence and arguments presented.
5. The cited references are relevant to the research and there is almost no self-citation.
6. The research gap is not identified.
7. English usage is appropriately not clear. Some Grammar errors in the introduction part.
8. Before the conclusion section please add managerial insight about the model.
9. Please add a table in the introduction to show a literature review of optimizing elliptic curve cryptography.
10. Please add a list of objectives and goals.
11. The authors should add a comparative analysis to show novelty and validity of the proposed approach.
12. Methodology of the paper is too weak and did not explain well. An algorithm/pseudo code must be given to get a better understanding of the framework.
Comments on the Quality of English LanguageMinor editing of English language required
Reviewer 3 Report
Comments and Suggestions for Authors
The authors propose an optimized SW implementation for Elliptic Curve Cryptography (Curve25519) on the ARM NEON architecture. They tested the implementation on multiple ARM devices demonstrating a speed-up of around 20%.
The paper is well written and organized, the results are promising and well presented. The State of the Art analysis is also good. I have only one remark. ECC is a well known cryptosystem, extensively studied and investigated. The acceleration of ECC on embedded platforms is important but providing secure implementation against Side-Channel Attacks (SCAs) is important too.
Maybe, the focus of this work is only related to the acceleration of the ECC and the authors did not puttheir effort on SCA. Nevetherless, the authors should at least mention the problem of the SCA in software and hardware implementations of ECC, and the fact that this problem exist but is out of the scope of this work. They can add a paragraph about SCAs, and cite some papers reporting attacks and countermeasures such as:
J. Fan, X. Guo, E. De Mulder, P. Schaumont, B. Preneel and I. Verbauwhede, "State-of-the-art of secure ECC implementations: a survey on known side-channel attacks and countermeasures," 2010 IEEE International Symposium on Hardware-Oriented Security and Trust (HOST), Anaheim, CA, USA, 2010, pp. 76-87, doi: 10.1109/HST.2010.5513110.
Zulberti, L.; Di Matteo, S.; Nannipieri, P.; Saponara, S.; Fanucci, L. A Script-Based Cycle-True Verification Framework to Speed-Up Hardware and Software Co-Design: Performance Evaluation on ECC Accelerator Use-Case. Electronics 2022, 11, 3704. https://doi.org/10.3390/electronics11223704
Abarzúa, Rodrigo, Claudio Valencia, and Julio López. "Survey for performance & security problems of passive side-channel attacks countermeasures in ECC." Cryptology ePrint Archive (2019).
Round 2
Reviewer 2 Report
Comments and Suggestions for Authors
The authors revised this paper according to the given suggestions.
This paper may be accepted.
Comments on the Quality of English LanguageModerate editing of English language required
Reviewer 3 Report
Comments and Suggestions for Authors
The quality of the paper was already good before the revision. It can be accepted as it is.